# Endothelial Dysfunction in Cardiorenal Conditions: Implications of Endothelial Glucocorticoid Receptor-Wnt Signaling

**DOI:** 10.3390/ijms241814261

**Published:** 2023-09-19

**Authors:** Mohammad Shohel Akhter, Julie Elizabeth Goodwin

**Affiliations:** 1Department of Pediatrics, Yale University School of Medicine, New Haven, CT 06511, USA; 2Vascular Biology and Therapeutics Program, Yale University School of Medicine, New Haven, CT 06511, USA; 3Department of Cellular and Molecular Physiology, Yale University School of Medicine, New Haven, CT 06511, USA

**Keywords:** endothelium, glucocorticoid receptor, cardiovascular disease, fibrosis

## Abstract

The endothelium constitutes the innermost lining of the blood vessels and controls blood fluidity, vessel permeability, platelet aggregation, and vascular tone. Endothelial dysfunction plays a key role in initiating a vascular inflammatory cascade and is the pivotal cause of various devastating diseases in multiple organs including the heart, lung, kidney, and brain. Glucocorticoids have traditionally been used to combat vascular inflammation. Endothelial cells express glucocorticoid receptors (GRs), and recent studies have demonstrated that endothelial GR negatively regulates vascular inflammation in different pathological conditions such as sepsis, diabetes, and atherosclerosis. Mechanistically, the anti-inflammatory effects of GR are mediated, in part, through the suppression of Wnt signaling. Moreover, GR modulates the fatty acid oxidation (FAO) pathway in endothelial cells and hence can influence FAO-mediated fibrosis in several organs including the kidneys. This review summarizes the relationship between GR and Wnt signaling in endothelial cells and the effects of the Wnt pathway in different cardiac and renal diseases. Available data suggest that GR plays a significant role in restoring endothelial integrity, and research on endothelial GR–Wnt interactions could facilitate the development of novel therapies for many cardiorenal conditions.

## 1. Introduction

The endothelium is a single layer of cells which constitutes the innermost cell lining of all blood vessels. This highly dynamic, semi-permeable barrier physically separates blood constituents from surrounding tissues. Although initially the endothelium was considered an inert barrier, it is now recognized as a unique organ with diverse physiological functions including maintenance of blood fluidity, vessel permeability, blood flow regulation, secretion of angiocrine factors [1], new blood vessel formation, leukocyte and platelet adhesion and aggregation, and transfer of oxygen and nutrients [2]. The adult human body contains approximately 10–60 trillion endothelial cells (ECs) representing a surface area of 400 m^2^ and a weight of approximately 1 kg [3]. The ubiquitous presence of the endothelium emphasizes its role in different physiological phenomena.

Research over the past decades has significantly enhanced our understanding of the molecular mechanisms underlying the formation and function of blood vessels and their endothelium. The importance of specific metabolic pathways used by ECs and the adaptations used to sustain their specialized functions have been recognized and stimulated great scientific interest. Several diseases are related to EC dysfunction, and evidence of the contributory role of metabolic maladaptation of the endothelium in such diseases is increasing. EC dysfunction is observed in various pathologies such as diabetes, cancer, and atherosclerosis, and the understanding of these associated pathologic mechanisms is crucial to identify potential targeted treatment strategies for clinical translation.

## 2. Cardiorenal Syndrome

Crosstalk between the heart and kidneys during the dysfunction of either or both organs have potential clinical implications for treatment strategies in acute and chronic conditions. These interactions underpin the pathophysiology of an emerging clinical condition, which is called cardiorenal syndrome (CRS). The term cardiorenal syndrome (CRS) was first described at a National Heart, Lung, and Blood Institute (NHLBI) Working Group conference in 2004. This earlier definition of CRS refers to the role of a diseased heart in causing kidney dysfunction. However, due to a greater understanding of CRS pathophysiology, it is now widely accepted that either the heart or the kidneys can be the primary source of this condition. Hence, a new definition of CRS restated this disease as a pathophysiologic condition of heart and kidneys in which the acute or chronic disorder of one organ may trigger acute or chronic dysfunction of the other organ [4]. CRS was divided into five sub-types by the 7th Acute Dialysis Quality Initiative Consensus Conference [5], which are detailed in Table 1.

## 3. Effects of Endothelial Dysfunction in Cardiorenal Syndrome

A single mechanism is not responsible for all cardiovascular disease nor for all causes of renal dysfunction. Nevertheless, one mechanism seems to be a pivotal commonality of many aspects of cardiorenal syndromes and stands out as the final common pathway: namely, endothelial dysfunction (ED). Over the past decade, ED has been recognized as a potential driver of the crosstalk between cardiac and renal dysfunction. The concept of ED as a link between cardiovascular disease and chronic kidney disease in CRS is reinforced by evidence from asymmetric dimethylarginine (ADMA)-mediated ED [6]. It has been reported that ED is associated with type 2 CRS [7], type 1 diabetes mellitus [8], and chronic CRS [9]. An inhibitor of nitric oxide synthase, ADMA, is a potential causative factor for ED in chronic kidney disease; hence, ADMA-mediated ED may contribute to the pathogenesis of CRS as a link between cardiovascular disease and chronic kidney disease in CRS patients [6]. An attractive hypothesis has been proposed to explain the possible involvement of ADMA in CRS [6]. During the interaction between kidney disease and cardiac disease, ADMA acts as a major factor which can induce ED by inhibiting endogenous nitric oxide synthase. ED induced by ADMA contributes to cardiovascular disease, and in contrast, this ED can, in turn, increase ADMA within the kidney. Hence, accumulated ADMA in chronic kidney disease is believed to be a missing link between cardiovascular disease and chronic kidney disease [6,10].

## 4. Glucocorticoid Receptor

Glucocorticoid receptor (GR) is a member of the nuclear receptor family encoded by the gene *NR3C1*, which is located on the short arm of chromosome 5 (5q31Y32) in humans. This receptor has four components: (1) N-terminal domain (NTD), (2) central DNA binding domain (DBD), (3) C-terminal ligand-binding domain (LBD), and (4) hinge region [11]. GR is a transcription factor and mediates the diverse functions of glucocorticoids. Glucocorticoids regulate an array of physiological functions including glucose metabolism, blood pressure, glomerular filtration rate (GFR) [12], immune response [13], and cognitive and emotional functions [14]. GR is ubiquitously expressed in most cell types including in endothelial cells, vascular smooth muscle cells (VSMC), and kidney cells, and it is conserved across species, underscoring its crucial role in various aspects of development, metabolism, inflammation, stress response, and homeostasis. Although this receptor is widely expressed, tissue-specific effects of glucocorticoids may exert a significant role on whole organism phenotypes [12].

## 5. Effects of GR on EC Metabolic Pathways

The expansion of the vascular network is initiated by changing metabolic demands during physiological organ growth to provide all tissues with adequate oxygen and nutrients. ECs remain quiescent in adults but retain the capability to rapidly initiate new vessel formation in response to stimulants or pathological conditions. Changes in cellular metabolism during angiogenesis are not merely coincidental; instead, they are a driving force for phenotypic differentiation [15]. A growing body of evidence suggests that different metabolic pathways such as glycolysis, fatty acid oxidation (FAO), and glutamine metabolism have significant roles in vessel formation. Metabolic reprogramming is associated with several disease conditions including cancer and diabetes. For instance, since glycolysis is increased in tumor ECs, lowering hyperglycolysis in tumors offers therapeutic benefits in preclinical tumor models. Similarly, any metabolic alteration in ECs potentiates the development of myofibroblast formation and proliferation, leading to kidney fibrosis in diabetes [16,17].

ECs produce 85% of their required ATP by glycolysis, whereas FAO contributes 5% of the total amount of ATP [15]. In addition to ATP production, FAO is associated with other cellular events in ECs, such as DNA synthesis [18], redox homeostasis [19], and epigenetic regulation [20]. Moreover, FAO is involved in endothelial-to-mesenchymal transition (EndMT) [21], which is a cellular trans-differentiation pathway through which ECs obtain mesenchymal characteristics, including loss of intercellular junctions and acquisition of a fibroblast-like morphology and expression of mesenchymal proteins (e.g., α-smooth muscle actin (α-SMA), fibronectin, and vimentin). These transformed mesenchymal cells are one of the major sources of myofibroblasts in different organs, including kidneys. In addition, EndMT induces pro-fibrogenic signaling in neighboring cells by autocrine and/or paracrine mechanisms. An aberration of FAO following cytokine stimulation triggers EndMT in human pulmonary microvascular endothelial cells (HPMVECs) [21] as well as in diabetic kidneys. This metabolic change is characterized by an early rapid decline in the level of carnitine palmitoyl transferase 1 (CPT1a), which is a rate-limiting enzyme of the FAO pathway. In contrast, the overexpression of CPT1a inhibits EndMT expression after cytokine treatment. The effects of endothelial GR on renal fibrosis have been studied in an experimental model of diabetic mice. The suppression of endothelial GR reduces FAO levels and facilitates cytokine reprogramming [22]. Conversely, FAO activation reduces kidney fibrosis by restoring endothelial GR expression in diabetic mice, suggesting the critical role of this receptor in FAO-mediated fibrosis regulation [22].

Interestingly, activation of the Wnt/β–catenin pathway can promote renal fibrosis, while its suppression inhibits myofibroblast activation and reduces the expression of fibroblast-specific protein 1, fibronectin, and type I collagen in the model of obstructive nephropathy [23]. Similarly, ECs collected from the kidneys of diabetic endothelial GR knockout (KO) mice showed an upregulation of Wnt signaling as well as fibrogenic markers such as α-SMA and fibronectin when compared to diabetic controls [22]. Overall, these findings demonstrate an intricate relationship among GR, Wnt signaling, and the FAO pathway in the context of renal fibrosis; however, further research is needed to delineate whether endothelial GR directly controls FAO in the mitochondria or indirectly regulates this process by regulating Wnt signaling.

## 6. Endothelial GR and Wnt Signaling

GR is present in the endothelium; however, its diverse functions in ECs are still being elucidated [12]. Endogenous glucocorticoids suppress local and systemic inflammation. Exogenous glucocorticoids such as dexamethasone are well-known anti-inflammatory agents, but the adverse effects of systemic glucocorticoids are common and sometimes can be severe.

Previous data show that endothelial GR suppresses vascular inflammation in a mouse model of sepsis. Mice lacking endothelial GR showed a higher expression of inflammatory cytokines and increased mortality and hemodynamic instability compared to endothelial GR-replete controls [24]. A similar phenotype was observed in human umbilical vein endothelial cells (HUVECs) when subjected to GR knockdown. These cells showed a higher expression of endothelial nitric oxide synthase (eNOS) and NF-κB activation after LPS treatment, indicating GR is a crucial regulator of NF-κB activation and nitric oxide synthesis in sepsis [24]. Moreover, endothelial GR negatively regulates atherogenesis in a mouse model in vivo [25]. Mechanistically, a loss of endothelial GR results in an upregulation of Wnt signaling both in vivo and in vitro [26], further demonstrating that Wnt signaling is a highly relevant pathway in vascular inflammation. Additional studies demonstrate that endothelial GR exhibits its anti-inflammatory effects, in part, by suppressing Wnt signaling in the vasculature [26].

## 7. Wnt Signaling in Cardiovascular Disease

The Wnt signaling pathway is an evolutionarily conserved developmental pathway that has been implicated in cellular communication in a wide array of physiological and developmental processes. Wnt proteins are mitogenic growth factors that comprise a family of 19 members in vertebrates. Wnt proteins are secreted by the cells where they act as ligands for various receptors, mainly the Frizzled (Fz) family of receptors, and different co-receptors, such as lipoprotein receptor-related protein 5 (LRP5), LRP6, Ryk or Ror2 [27]. It is well established that Wnt ligands transmit their signal via both canonical (β-catenin-dependent) and non-canonical (β-catenin-independent) pathways. The canonical pathway is also referred as the Wnt/β-catenin pathway, and the non-canonical pathway includes the Wnt–planar cell polarity pathway (Wnt–PCP pathway) and the Wnt–calcium pathway [28]. Based on its functions, the Wnt family ligands are divided into two categories: Wnt1 and Wnt5a. The Wnt1 category is composed of Wnt1, Wnt2, Wnt2b, Wnt3, Wnt3a, Wnt7a, Wnt8, Wnt8b, and Wnt10a, which are involved in the canonical signaling pathway, while Wnt5a category includes Wnt4, Wnt5a, and Wnt11 and activates the non-canonical signaling pathway [28]. Wnt family proteins are essential for different aspects of cardiac, renal, and vascular development including myocardial precursor cell specification, cardiac valve formation, and early nephrogenesis as well as VSMC proliferation. Wnt signaling is mostly inactive in healthy adult organs, but its reactivation is generally observed in diverse disease conditions. An increasing body of evidence suggests the relevance of post-natal Wnt signaling in different disease conditions, such as chronic kidney disease, atherosclerosis, diabetes, cardiovascular disease, and inflammation [29]. 

Cardiovascular disease is a significant health problem and remains the leading cause of mortality worldwide [30]. Wnt signaling plays a critical role in the progression of heart disease in terms of both metabolic alterations (i.e., insulin sensitivity) and cardiovascular remodeling and structural changes such as fibrosis, sclerosis, hypertrophy, and smooth muscle cell proliferation [31]. The effects of β-catenin on cardiac hypertrophy have been investigated by several groups, although the full spectrum of these effects remains to be fully elucidated [32]. β-catenin protein levels are higher in hypertrophic hearts compared to controls [33], and this overexpression can induce different hypertrophic markers, including β-myosin heavy chain (β-MHC) and α-actin, activate c-Myc and Snail1 transcription factors, and enhance atrial natriuretic peptide (ANP) and brain natriuretic peptide (BNP) expression, leading to a hypertrophic growth of cardiomyocytes and cardiac dysfunction [34,35]. The pharmacologic inhibition of Wnt/β-catenin signaling or cardiomyocyte-specific deletion of β-catenin attenuates the hypertrophic response of cardiomyocytes in vivo [35,36,37]. 

### 7.1. Wnt Signaling in Atherosclerosis

Atherosclerosis is a chronic inflammatory disease of the vascular wall that forms the pathological basis of a range of cardiovascular complications, including stroke and myocardial infarction. It is characterized by the complex phenotypic change in endothelial cells, VSMC, and macrophages, leading to the deposition of fibrous elements, lipids, and inflammatory cells within the arterial wall [38,39]. Consequently, the affected arteries either become narrow such as in angina pectoris, or they rupture, which may result in arterial thrombotic occlusion.

ED, increased thickening of the intima, inflammation, and vascular calcification are the pivotal features of atherosclerosis. ED is the first step in the pathogenesis of atherosclerosis. It causes a higher expression of adhesion molecules including intercellular cell adhesion molecule 1 (ICAM1) and vascular cell adhesion molecule 1 (VCAM1) and releases proinflammatory cytokines, resulting in monocyte, lymphocyte, and platelet adhesion to the endothelium [40]. Moreover, ED engenders endothelial hyperpermeability, permitting an intraplaque invasion of monocytes and macromolecules such as low-density lipoprotein (LDL) [41]. Nitric oxide (NO) prohibits VSMC migration in physiological conditions; however, its levels are reduced in ED, resulting in the proliferation and migration of VSMC into the intima that contributes to intimal thickening [42]. The contribution of Wnt signaling in ED has been investigated in several studies [43,44]. Increased Wnt5a expression has been observed in the macrophage-rich regions in both human and murine atherosclerotic lesions [45]. It has also been shown that Wnt5a increases endothelial inflammation in the progression of atherosclerosis by inducing expression of the inflammatory gene cyclooxygenase-2 (COX-2) as well as other cytokines including interleukin (IL)-1a, IL-5, IL-6, and IL-8 [46]. A pulse-like treatment of Wnt5a enhances COX-2 expression even more compared to continuous treatment. Wnt5a activates the NF-κB pathway and calcium chelators and protein kinase C inhibitors suppress Wnt5a-induced activation, suggesting a role of Wnt/Ca^2+^/protein kinase C pathway in endothelial inflammation [46]. The role of canonical Wnt signaling in the expression of the redox regulatory protein p66^shc^ has been studied in HUVECs. The phosphorylation of p66^shc^ at Ser36 increases oxidative stress and thus promotes ED and atherosclerosis [47,48]. The canonical Wnt ligand Wnt3a induces c-Jun N-terminal Kinase (JNK)-mediated phosphorylation and the activation of p66^shc^ in endothelial cells. Active p66^shc^ dephosphorylates β-catenin and induces β-catenin-dependent oxidative stress, resulting in impaired vasorelaxation [49]. However, an inhibition of Wnt signaling with the Wnt ligand antagonist Dickkopf-1 (Dkk1) suppresses Wnt3a-mediated p66^shc^ phosphorylation. These results demonstrate a crucial role of p66^shc^ in Wnt3a-stimulated endothelial oxidative stress and dysfunction [49]. 

EndMT facilitates the initiation and progression of atherosclerosis. Interestingly, Dkk-1 has been reported to enhance EndMT by downregulating platelet endothelial cell adhesion molecule (PECAM), vascular endothelial cadherin (VECAD), and claudin 5 in aortic ECs [50,51]. It also induces apoptosis in ECs by inducing endoplasmic reticulum (ER) stress [52]. Clinical studies in acute coronary syndrome (ACS) patients demonstrate a positive correlation between Dkk-1 plasma levels and the severity of coronary atherosclerosis [53]. A strong Dkk-1 immunoreactivity was observed in platelet aggregates at the site of plaque rupture. Moreover, in vitro experiments demonstrate that both platelet- and endothelial-derived Dkk-1 contributes to platelet-dependent endothelial activation, leading to an enhanced release of inflammatory cytokines. The inflammatory effects of Dkk-1 involved activation of the NF-κB pathway and suppression of the Wnt/β-catenin pathway [54]. These findings suggest that Dkk-1 secreted from activated platelets within the inflammatory microenvironment triggers endothelial dysfunction and participates in atherosclerotic plaque formation. Dkk-1 is also associated with the activation of β-catenin independent-Wnt signaling, although the mechanistic details are not well understood. Hence, Dkk-1 has been hypothesized to be involved in shifting the Wnt signaling balance from the β-catenin-dependent to the β-catenin-independent pathway [55]. However, the role of Dkks in the pathophysiology of the arterial wall is only partially elucidated, and further studies are needed to more fully understand their roles in atherosclerotic-related diseases [55].

Macrophages play a crucial role in the progression of atherosclerosis. Macrophages contain lipids that form foam cells, which are a characteristic of atherogenic lesions. Foam cells express collagenase in response to inflammatory mediators which weaken the fibrous cap, leading to plaque rupture and acute atherothrombotic complications. In addition, they secrete cytokines and growth factors that promote migration to, and the proliferation of, VSMCs to the growing plaque [56]. An increased level of Wnt5a has been observed in macrophage-rich areas of atherosclerotic plaques [45]. Moreover, oxidized-low density lipoprotein (ox-LDL) plays a crucial role in atherogenesis. The expression of Wnt5a is related to the severity of atherosclerotic lesions, and ox-LDL significantly induces the mRNA expression of *Wnt5a* in human monocyte-derived macrophages [57]. Moreover, activation of the canonical Wnt signaling pathway has been reported in plaque macrophages during atherosclerosis regression [58]. Interestingly, another study suggests an inverse relationship between Wnt signaling and the severity of atherosclerosis. IL-4 is required for atherosclerosis resolution. Wnt3a induces prostaglandin E2 (PGE2), which enhances the responsiveness of macrophages to IL-4 and thus may promote the resolution of atherosclerosis [59]. Wnt signaling is also involved in vascular calcification, which is a crucial marker for atherosclerosis [60]. Data suggest that Wnt signaling regulates multiple aspects of both atherosclerotic lesion formation and regression. A more integrated research approach, especially of tissue-specific Wnt modulation, is warranted to explain some discrepancies about the role of Wnt among different studies.

### 7.2. Wnt Signaling in Myocardial Infarction

Myocardial infarction (MI) remains a significant cause of mortality worldwide. It is caused by the acute occlusion of one or more coronary arteries, resulting in myocardial necrosis in the affected regions due to a lack of oxygen and nutrients. MI usually induces oxidative stress and evokes inflammatory responses, which are followed by the transformation of cardiac fibroblasts to cardiac myofibroblasts [61]. Wnt signaling pathways exert a significant role in heart development but are also active in the post-MI adult heart [62]. Despite the availability of several pharmacologic interventions such as calcium channel blockers and renin–angiotensin–aldosterone (RAAS) inhibitors, the progression and pathological remodeling of MI remain irreversible. Studies about the protective effects of the Wnt pathway inhibitors in cardiac diseases have stimulated interest in the possibility of employing these inhibitors to treat MI [62,63]. 

The pathological process of MI has three major phases: (i) inflammatory reaction, (ii) the formation of granulation tissue, and (iii) fibrosis [62]. The expression of several Wnt proteins, including Wnt-2, Wnt-4, Wnt-10b, and Wnt-11, are induced as soon as 5 days after MI [64]. Another study reported enhanced levels of Wnt-1 from 1 to 14 days after MI and upregulated Wnt-4 expression from 7 to 14 days after MI [65]. *Axin2* is one of the target genes of Wnt signaling, and Axin2 promoter-driven LacZ expression has been observed in cardiomyocytes located at the infarct border zone [66]. Additionally, Wnt signaling was also found to be increased in ECs, fibroblasts, and progenitor cells in at 7- and 14-days post-MI [66]. 

The inflammatory reaction is triggered in response to MI, which is important for removing necrotic debris and postinfarction repair; however, excessive inflammation may cause myocardial remodeling and heart failure. A specific Wnt protein, Wnt-5a, which is exclusively expressed in cardiomyocytes [67], can promote the release of different proinflammatory cytokines, such as IL-1, IL-6 and IL-8 from mononuclear cells, demonstrating an involvement of Wnt signaling in inflammation [68].

Finally, macrophages in the infarct area engulf necrotic myocardial cells and secrete cytokines and chemokines. Although Wnt/β-catenin signaling is inactive in normal cardiac macrophages, increased levels of β-catenin have been observed in cardiac macrophages, especially Ly6C+ proinflammatory macrophages post-MI [69]. Activated β-catenin promotes macrophage-mediated inflammatory responses by inducing the expression of IL-1β, IL-6, and tumor necrosis factor-α (TNF-α). Moreover, β-catenin may induce inflammation in MI by stimulating the NF-κB pathway. Both the Wnt/β-catenin pathway and NF-κB were shown to be activated after MI [70]. β-catenin facilitates the expression of NF-κB target gene expression by binding the β-catenin-TCF/LEF complex to the promoters of NF-κB target genes. β-catenin overexpression in cardiomyocytes causes enhanced cytokine release and activation and a nuclear localization of NF-κB. This suggests that β-catenin promotes inflammation by activating the NF-κB pathway [71]. In addition, glucocorticoids negatively regulate NF-κB signaling and hence have been used for the treatment of conditions mediated by NF-κB. The regulation of NF-κB by glucocorticoids can be explained by two hypotheses: (1) glucocorticoids increase the expression of IκB, a suppressor NF-κB pathway, or (2) they directly interact with NF-κB [24].

## 8. Wnt Signaling in Diabetic Nephropathy

Diabetic nephropathy (DN) is a chronic kidney disease that results from uncontrolled diabetes for a prolonged period, generally years. It is a major complication of diabetes and often leads to end-stage renal failure. Clinically, DN is characterized by proteinuria, a reduction in glomerular filtration rate, hypertension, and eventual renal failure. The major features of DN include thickening of the basement membrane, podocyte damage, mesangial expansion due to accumulation of the extracellular matrix (ECM), and glomerular and tubular cell damage, which lead to interstitial fibrosis and glomerular sclerosis [72]. 

Different signaling pathways, such as transforming growth factor β (TGF-β)/Smad pathway, PI3K/Akt, p38 mitogen-activated protein kinase (MAPK), and Janus kinase/signal transducer and activator of transcription (JAK/STAT) signaling have been implicated in the pathogenesis of DN [73]. Moreover, substantial literature has reported the critical role of Wnt signaling in diabetic renal injury, especially to mesangial cells (MCs), podocytes, and tubular epithelial cells [72,74,75]. This signaling has also been associated with renal tubulointerstitial fibrosis and glomerular sclerosis, both of which are pathological hallmarks of DN [76]. The canonical Wnt signaling pathway is relatively inactive in normal adult kidneys; however, it is activated during kidney injury [77]. In addition to the Wnt canonical pathway, non-canonical Wnt signaling has begun to attract attention as a potential mechanism for the pathogenesis of DN [74].

Podocytes are terminally differentiated epithelial cells that cover the outer layer of the glomerular basement membrane. Podocytes are crucial for regulating glomerular functions, including the formation and renewal of the basement membrane and maintaining the structural stability of the capillary loop. The glomerular filtration barrier consists of three layers: the fenestrated endothelium, the podocytes, and the glomerular basement membrane. It controls the influx and efflux of proteins [78], and hence, podocyte damage and dysfunction inevitably contribute to the pathogenesis of DN.

β-catenin expression is necessary for podocyte differentiation from renal vesicles, but an abnormal upregulation of β-catenin may lead to podocyte dysfunction and damage [75]. Uncontrolled activation of the Wnt/β-catenin pathway inhibits Wilm’s tumor 1 (WT1)-mediated gene expression and facilitates podocyte de-differentiation and mesenchymal transformation. Conversely, blocking the Wnt/β-catenin pathway restores WT1-mediated gene silencing, suggesting the role of this pathway in podocytes integrity [79]. Wnt/β-catenin signaling is also upregulated in podocytes in humans with DN as well as in streptozotocin (STZ)-induced diabetic mice [80]. Different Wnt proteins, such as Wnt1, Wnt2B, Wnt4, Wnt6, and Wnt16, have been shown to be upregulated in experimental DN. While Wnt/β-catenin activation results in podocyte dysfunction, the suppression of Wnt signaling by Dkk1 restores podocyte function and reduces albuminuria. However, the deletion of β-catenin in podocytes and overexpression of Dkk1 both enhance the severity of STZ-induced DN, suggesting that both the hypoactivation and hyperactivation of Wnt signaling may promote renal injury during DN [81]. Table 2 summarizes the involvement of Wnt signaling in different cardiovascular and renal diseases.

## 9. Wnt Signaling in Cardiorenal Syndrome

The potential role of Wnt/β-catenin signaling has been investigated in the context of CRS2, which is a disease condition in which chronic abnormalities in cardiac function cause progressive chronic kidney disease. Zhao et al. reported an upregulation of Wnt/β-catenin signaling and RAAS activation in the heart after 8 weeks of transverse aortic constriction (TAC) surgery in mice. RAAS activation triggers proinflammatory cytokine release, especially TNFα, which, in turn, activates β-catenin in the kidney [82]. The activation of β-catenin is linked to the onset and progression of chronic kidney disease [83]. The pharmacologic inhibition of β-catenin ameliorates cardiac injury, restores heart function, and mitigates the observed renal lesions, demonstrating Wnt/β-catenin signaling as a unified pathogenic mediator of both kidney and heart injury in a mouse model of CRS2 [82].

## 10. Conclusions

Dysfunctional endothelium is the central insult which initiates the pathological progression of a wide range of diseases such as cardiovascular disease, renal disease, acute respiratory distress syndrome (ARDS), Alzheimer’s disease, and glaucoma. Comprehensive investigations targeting this common pathway may help in the development of novel therapeutic interventions to counteract the pathogenesis of corresponding diseases. Vascular endothelium expresses GR, and tissue-specific signaling of this receptor has profound implications in renal and cardiovascular disease. The negative feedback loop between endothelial GR and Wnt signaling has been reported recently, indicating an additional signaling pathway, other than NF-κB, a well-known target of GR, that modulates inflammation both in vitro and in vivo [26]. Wnt signaling is associated with many different cardiovascular and renal diseases (Figure 1). Hence, an improved understanding of the GR–Wnt signaling axis in regulating endothelial dysfunction will not only permit the development of novel therapeutic strategies but may also reduce the off-target side effects of glucocorticoids. Ongoing studies of this interaction in ECs, as well as other cell types, have the potential to catalyze new therapies for many common cardiorenal conditions.

## Figures and Tables

**Figure 1 ijms-24-14261-f001:**
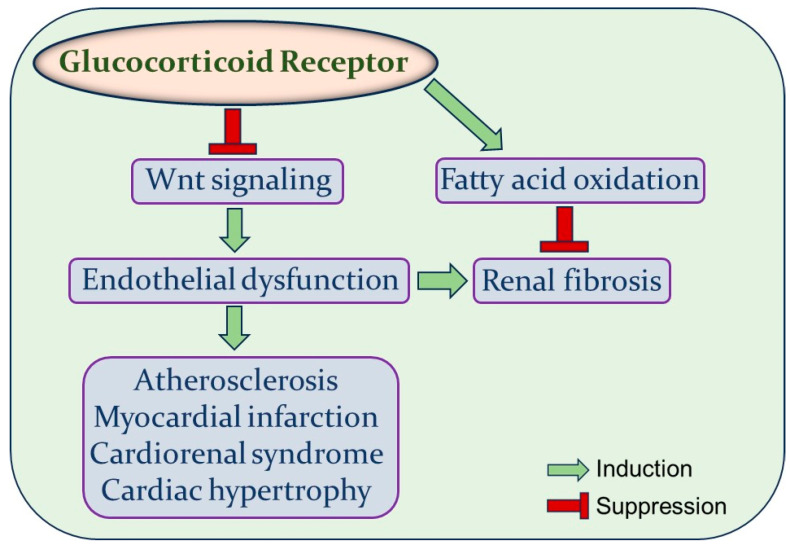
Graphical representation of the effects of endothelial GR–Wnt interaction in cardiovascular and renal disease. Glucocorticoid receptors suppress Wnt signaling in endothelial cells. Consequently, Wnt pathways participate in the pathogenesis of several cardiovascular and renal disease by inducing endothelial dysfunction. Moreover, disruption of the FAO pathway in endothelial cells contributes to renal fibrosis through EndMT activation, while endothelial GR counteracts renal fibrosis by restoring FAO levels.

**Table 1 ijms-24-14261-t001:** Definition and classification of cardiorenal syndrome.

Phenotype	Nomenclature	Definition
Type 1 CRS	Acute CRS	Acute worsening of cardiac function leading to renal dysfunction.
Type 2 CRS	Chronic CRS	Chronic abnormalities in cardiac function leading to renal dysfunction.
Type 3 CRS	Acute reno- cardiac syndrome	Acute worsening of renal function causing cardiac dysfunction.
Type 4 CRS	Chronic reno- cardiac syndrome	Chronic abnormalities in renal function leading to cardiac disease.
Type 5 CRS	Secondary CRS	Systemic conditions causing simultaneous dysfunction of the heart and kidney.

**Table 2 ijms-24-14261-t002:** Wnt signaling in cardiovascular and renal complications.

Cell Types	Modulators	Molecular Pathways	Effects
Cardiomyocytes and cardiac fibroblasts	Angiotensin II	↑ Wnt/β-catenin	Cardiac hypertrophy and myocardial fibrosis [35]
Endothelial cells	Wnt5a	↑ Non-canonical Wnt pathway ↑ NF-κB pathway	Endothelial inflammation and atherosclerosis [46]
HUVECs	Wnt3a	↑ Wnt/β-catenin	Endothelium oxidative stress and dysfunction [49]
Platelets and HUVECs	DKK-1	↑ NF-κB pathway ↓ Wnt/β-catenin	Endothelial dysfunction and atherosclerotic plaque formation [54]
Macrophages	ox-LDL	↑ Non-canonical Wnt pathway	Expression of Wnt5a and progression of atherosclerosis [57]
Cardiac macrophages	Post MI	↑ Wntβ-catenin ↑ NF-κB pathway	Macrophage-mediated inflammatory response after MI [69,70]
Podocytes	Adriamycin	↑ Wnt/β-catenin	Podocyte dysfunction and albuminuria [80]

## Data Availability

Data are available from the corresponding author upon reasonable request.

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
