# Peer review of "Endothelial Dysfunction in Cardiorenal Conditions: Implications of Endothelial Glucocorticoid Receptor-Wnt Signaling"

_ijms, 2023, doi:10.3390/ijms241814261_

Round 1

Reviewer 1 Report

The paper presented for review addresses the importance of endothelial glucocorticoid-Wnt receptor signaling in endothelial dysfunction in cardiorenal conditions.

The manuscript is written in a logical, coherent manner, and includes a review of a broad literature database.

My minor suggestions are as follows:

line 89 - it would be worth to give some examples of the functions of glucocorticoids briefly

since the work presents often complex relationships and links between various factors and mechanisms, perhaps it would be worth considering the possibility of graphical presentation of some of them

Reviewer 2 Report

This paper mainly reviews the role of Wnt and its signaling, mainly through beta-catenin, in endothelial dysfunction and the cardiorenal syndrome.

Although it is a good synthesis on a vast field of research, I believe some adjustments need to be made to reach a larger readership. Particularly, I was a bit disappointed by the presentation of the molecular aspects of the review.  What is the Wnt family? What are the main players of Wnt signaling in endothelial cells? What are the main players of Wnt canonical and non-canonical pathways? What is the importance of the beta-catenin pathway and/or why the authors focused so much on this pathway and so little on cyclin D1 or c-Myc for instance? Also, given the title, I found little on endothelial GR and Wnt signaling, I was expecting to read more in this section.

Minor comments:

I found hard to follow the presentation of the impact of Dkk1 in EndMT (lines 213-226) and somewhat contradictory. Could the authors clarify?

The different sections have little links between one and the other and the logic of the flow/structure seems to be missing.

Figure 1 is rather poor with respect to the content and readership. Maybe good to add a second more “molecular” figure.

English quality is good with only a few minor corrections to be made.

Reviewer 3 Report

The manuscript is very well written and provides a thorough overview of the Wnt signaling in ED for a broad range of pathophysiology. Authors may discuss ED and its relation to pericytes in different cardiorenal conditions. I have one suggestion to include a table that summarizes the key points including the cell types (column 2), molecular pathways (Column 3) affected in the various disease models (Column3) discussed in the review.

English is well written. I could not find any errors. 
